# Evolution of Medical Modeling and 3D Printing in Microvascular Midface Reconstruction: Literature Review and Experience at MD Anderson Cancer Center

**DOI:** 10.3390/medicina59101762

**Published:** 2023-10-02

**Authors:** John W. Shuck, Rene D. Largo, Matthew M. Hanasono, Edward I. Chang

**Affiliations:** Department of Plastic Surgery, University of Texas MD Anderson Cancer Center, Houston, TX 77030, USA

**Keywords:** virtual surgical planning, medical modeling, microvascular midface reconstruction, maxillectomy reconstruction

## Abstract

Reconstruction of the midface represents a challenge for reconstructive microsurgeons given the formidable task of restoring both aesthetics and functionality. In particular, preservation of proper globe positioning and maintaining normal vision are as important as restoring the proper projection of the midface and enabling a patient to speak and eat as normally as possible. The introduction of virtual surgical planning (VSP) and medical modeling has revolutionized bony reconstruction of the craniofacial skeleton; however, the overwhelming majority of studies have focused on mandibular reconstruction. Here, we introduce some novel advances in utilizing VSP for bony reconstruction of the midface. The present review aims (1) to provide a review of the literature on the use of VSP in midface reconstruction and (2) to provide some insights from the authors’ early experience.

## 1. Introduction

Virtual surgical planning (VSP) has grown tremendously over the years since its introduction into craniofacial surgery [1,2]. This technology has allowed for more precise planning of osteotomies and superior realignment that theoretically improves bony union and restores the proper aesthetics of the craniofacial skeleton. This technology has been used widely in the oncologic as well as the trauma setting, and its use has rapidly expanded to the point that it has become the standard of care at many institutions. The overwhelming majority of studies demonstrating improved outcomes and benefits with VSP have focused on reconstruction of the mandible [3,4,5]; however, its role has expanded into reconstruction of the midface as well [6,7,8].

The use of VSP for reconstruction of the maxilla is a logical extension applying the same technology used in mandible reconstruction to the midface. Given the standardization of imaging of the craniofacial skeleton and donor sites at most high-volume institutions, VSP is growing in popularity for midface reconstruction as well [9,10,11,12]. The fibula osteocutaneous free flap remains the workhorse flap for bony reconstruction following resection of head and neck cancers at our institution, and a CT angiogram is often obtained during the patient’s preoperative evaluation to aid not only in delineating the vascular anatomy of the lower extremity, but also designing customized cutting guides and implants [13,14]. While the workflow and utilization of VSP have evolved over time, we favor designing customized cutting guides with 3D-printed titanium plates and potentially even placement of immediate dental implants for mandibular reconstruction [15,16]. With our experience in mandible reconstruction, we extrapolated our knowledge to midface reconstruction and present our surgical approach and technique incorporating the latest advances in computer-assisted design and modeling using virtual surgical planning. The present study aims to provide a review of the literature using VSP in midface reconstruction and to present some novel insights from the authors’ experience.

### 1.1. Preoperative Evaluation

All patients undergoing tumor extirpation and midface reconstruction require routine imaging prior to surgery; however, the standard work-up is no different than that for any other patient. Every patient should undergo a thorough history and physical, paying particular attention to the prior history of surgery and treatment, such as radiation, which will impact considerations for reconstruction. Further, if patients have had prior surgery, especially a previous free flap, the evaluation of donor sites and recipient vessels is critical prior to proceeding with surgery.

For bony reconstruction of the midface, proper imaging is necessary to plan the resection and reconstruction using virtual surgical planning. In general, fine-cut CT scans of the head and neck are needed to plan the resection, so the cuts ideally should be no larger than 1.25 mm within one month of surgery. Imaging of the head and neck not only aids in determining the extent of disease and planning of the resection, but also affords the opportunity to evaluate the availability of recipient vessels. In addition, imaging allows the surgeon to plan the reconstruction to determine points of fixation for the flap as well as restoration of dental occlusion, midface projection, and potentially the position of the orbit and globe.

As the fibula flap has largely become the primary donor site for bony reconstruction, a CT angiogram is also routinely obtained during the preoperative work-up [14]. Historically, the authors have only obtained an angiogram when a patient had an abnormal peripheral vascular exam or a significant history of peripheral vascular disease. In the current era of VSP, imaging of the lower extremity introduces a number of other advantages that were previously not considered. The angiogram component demonstrates the presence of stenosis and large-vessel atherosclerosis or aberrant anatomy such as peronea magna, while advances in imaging can also now provide information regarding the perforators arising from the peroneal pedicle to the overlying skin. This allows for precise planning of the skin paddle in juxtaposition to the bony osteotomies. Obviously, the scans allow for the design of custom cutting guides, but they also potentially allow for the placement of immediate dental implants, which have demonstrated promising early outcomes at other institutions [17,18].

### 1.2. Customized Titanium Plates

The evolution of VSP has been closely linked to advances in the production of titanium plates for microvascular head and neck reconstruction. In mandible reconstruction, professionally bent reconstruction bars were replaced with milled titanium plates, which have now been replaced with 3D-printed titanium plates. In the arena of midface reconstruction, the contouring of the fibula often requires multiple segments and therefore multiple osteotomies [19]. VSP technology provides cutting guides to restore the proper dimensions of the maxilla, but the earliest generation only provided models based on preoperative imaging. Historically, mini-plates represented the mainstay for bony fixation; however, in the modern era of VSP, careful planning should allow two weeks of lead time to allow sufficient time for the production of 3D-printed titanium plates. The current generation of 3D-printed plates allows for the design of intricate plates based on the precise unique configurations of the fibula needed to restore the contour and profile of the patient’s craniofacial skeleton. The newest production technique also reduces the hardware burden and produces engineered plates with a dramatic reduction in the plate profile, which will hopefully translate into fewer plate-related complications.

Aside from the 3D-printed plates used for bony fixation, the same technological advances have also ushered in custom-designed, patient-specific plates for orbital reconstruction (Figure 1). When the orbital floor is sacrificed, re-establishing the proper position of the globe and orbital volume is critical for preserving normal vision. In the past, a model would be produced of the patient’s skull, and a generic titanium orbital floor plate would be contoured based on the model. However, again in the modern era of VSP, a patient-specific orbital floor plate can now be 3D-printed to the exact dimensions of the native orbital floor [20]. Early experience with this technology has demonstrated remarkable improvements in restoring proper globe position and preserving normal vision. Despite the advantages of 3D-printed orbital floor plates, consideration again must be given to proper planning with appropriate lead time as well as post-operative hardware-related complications like infection or extrusion [20,21].

### 1.3. Intraoperative Technique

The free fibula osteocutaneous flap represents our primary option for maxillary reconstruction. In general, we prefer to harvest the ipsilateral leg, allowing placement of the skin paddle intraorally for reconstruction of the palate. In situations where additional soft tissue is needed for obliteration of the maxillary sinus or to provide additional soft tissue coverage of the hardware, a pedicle flexor hallucis longus (FHL) or a portion of the soleus muscle can be taken with the fibula flap, and in certain circumstances, a second skin paddle may also be possible if multiple perforators exist [22]. In many circumstances, a full neck dissection is not warranted, so the facial vessels at the level of the mandible can be reliably used. Alternatively, the superficial temporal vessels can also be considered, but this is a secondary choice given the smaller-caliber vessels and high propensity for vasospasm.

Once the osteotomies are completed, plating is performed in the leg to limit the ischemia time; however, rendering the fibula ischemic for the osteotomies and plating is also an option as the custom cutting guides reduce ischemia time. Once the fibula construct has been secured to the titanium plate, the flap is transferred to the head and neck for bony fixation. We prefer to inset the skin paddle into the soft tissue defect prior to performing the microvascular anastomosis. The additional soft tissue components can be placed into the maxillary sinus, or the second skin paddle can be de-epithelialized to provide additional coverage of the hardware or to fill the dead space.

In the setting of orbital floor reconstruction, early iterations using VSP only provided a 3D model of the patient’s craniofacial skeleton, and a titanium plate would be contoured to the model in order to match the patient’s native orbital floor as closely as possible. The titanium plate would then be secured to the remaining zygoma to provide proper support for the globe prior to fixation of the fibula flap. When only a soft tissue flap is needed, coverage of the orbital plate with vascularized tissue is critical for preventing post-operative complications. An anterolateral thigh (ALT) flap can often provide ample soft tissue to close a palatal defect, fill the dead space of the maxillary sinus, and also provide vascularized fascia for coverage of the orbital floor plate [23]. Using the most advanced VSP technology, a custom, patient-specific titanium plate can be printed in any configuration to restore the precise position of the orbital floor to maintain the exact orbital volume following the resection. The printing can create a plate with complex extensions for bony fixation to the remaining bony pillars or to the fibula flap.

### 1.4. Free Flap Donor Site Selection

The authors prefer to use the free fibula flap for the majority of maxillectomy defects. With the increased understanding of perforator flaps and microvascular surgery, the skin paddle of the fibula flap can not only be harvested with reliability and success, but multiple skin islands can also be designed when more soft tissue is needed [22]. Since the fibula can tolerate multiple osteotomies, a single fibula can provide ample bone length to reconstruct a total maxillectomy defect. Aside from the length, the fibula also has sufficient bone stock, which offers the opportunity to support dental implants. With proper planning and patient selection, dental implants can be placed at the time of the primary reconstruction. The patient-specific cutting guides can be engineered to incorporate predictive screw holes for the custom-printed titanium plate so the screws will not interfere with the predictive drill holes for the dental implants. This may afford patients who require adjuvant radiotherapy the potential for complete dental rehabilitation and would otherwise not be candidates for secondary dental implants following radiation.

However, when the fibula is not available, and for certain types of defects, alternative osteocutaneous flaps should also be considered. While the overwhelming majority of cases are reconstructed with a fibula, VSP technology can also be used with high reliability for other flaps, such as the scapula or iliac crest [24,25,26,27]. A more recent advance in bony reconstruction of the midface was the introduction of the medial femoral condyle (MFC) flap for the reconstruction of midface defects [28]. The use of vascularized bone is recommended in the setting of previous radiation or anticipated adjuvant radiation, which can have a significant detrimental impact on the long-term outcomes. Given the direct communication with the aerodigestive system, placing even the most advanced engineered plate into a contaminated space with oronasal flora always introduces the potential for infection and need for removal of the hardware [29,30].

### 1.5. Alternative Vascularized Bone Flaps

While the fibula flap remains the workhorse flap for bony reconstruction of the midface and maxilla, certain defects do not require a fibula flap. Under these circumstances, other bony options can be considered and included in the armamentarium of bone flaps. Defects involving only a portion of the hard palate or perhaps the orbital floor may be reconstructed with soft tissue or perhaps a titanium plate, respectively. However, there are concerns regarding the use of hardware in a contaminated field that will likely require radiation, which may lead to a high risk of infection. For these reasons, we have incorporated a variety of osteocutaneous flaps in our approach for midface reconstruction.

The scapula represents an alternative option for bony reconstruction and can provide both vascularized bone as well as soft tissue [31,32]. A portion of the latissimus dorsi muscle, a skin paddle based off a thoracodorsal artery perforator, or a parascapular skin paddle, can be harvested to create a chimeric flap, allowing for complex reconstruction of midface defects (Figure 2). With the advances in medical modeling, the technology routinely employed for fibula-flap-based maxillectomy reconstruction can also be used for engineering an osteocutaneous flap using the scapula donor site. Similar to the use of medical modeling for immediate dental implants in fibula flaps, immediate dental implants can also be incorporated into scapula flap reconstruction (Figure 3).

Aside from the fibula and scapula donor sites, we also investigated the use of the medial femoral condyle (MFC) as a potential option for midface and maxillectomy reconstruction [33,34]. While it is most commonly employed for the reconstruction of extremity bone defects, the MFC also provides reliable vascular bone that can be utilized for head and neck reconstruction as well. Given the recent innovation of using the MFC for midface reconstruction, the use of medical modeling has not yet been incorporated into the workflow when planning an MFC flap. However, with increased experience in using the MFC in head and neck reconstruction, the authors now favor using vascularized bone when there is an anticipated need for radiation. Despite the high fidelity of customized 3D-printed titanium plates, the risks of hardware infection or extrusion cannot be underestimated. Using vascularized bone flaps may offer an alternative solution to 3D-printed titanium plates.

However, one important factor that must be considered is the limited pedicle length of the MFC flap. While a vein graft is one option, the authors favor the creation of an engineered chimeric flap by coupling the MFC flap to a second soft tissue flap in a flow-through orientation. The soft tissue flap, such as an ALT flap, must have an adequate pedicle length to reach either the facial vessels or the superficial temporal vessels without the need for vein grafting (Figure 4). The decision for increased operative time and a second donor site versus the increased risk of complications with the use of vein grafts is at the discretion of the reconstructive microsurgeon [35].

## 2. Discussion

Microvascular reconstruction of the midface has witnessed tremendous advances over time, from generic titanium plates and bone grafts to microvascular osteocutaneous flaps incorporating cutting-edge technology with preoperative imaging and computer-assisted design and modeling [36]. The growth in virtual surgical planning technology has revolutionized reconstruction of the craniofacial skeleton in congenital pediatric cases as well as traumatic defects [37,38,39,40]. The present review offers a summary of the most up-to-date studies in the literature, and also offers some insights from the authors’ early experience in oncologic reconstruction, where extensive resections are performed in order to obtain the highest chance for cure. However, in the setting of oncologic resections, reconstruction is not simply aimed at replacing the tissue removed; the surgery must also consider the impact of radiation therapy on the final outcome.

The use of VSP has become the standard of care for mandibular reconstruction in many institutions [41,42,43]. Early iterations of this technology simply provided models and professionally contoured reconstructions bars; however, as the technology evolved, models allowed the design of customized cutting guides and titanium plates. Even titanium plates have witnessed tremendous advances, from professionally contoured plates to milled titanium plates to the newest generation of 3D-printed titanium plates. The new 3D-printed plates allow for the most intricate designs that can be engineered for any defect and still permit stable bony fixation to the remaining bony skeleton [44]. However, the technology is not without limitations that should be considered by the reconstructive surgeon.

Despite the potential advantages that VSP provides in increasing operative efficiency and decreasing operative times, there are limitations and costs associated with its use that need to be considered. However, with the reduced operative time and ischemia time, the costs of the actual guides and plates may be justified [45,46]. There are a number of other important considerations when using VSP. Aside from the obvious costs associated with the customized cutting guides and 3D-printed plates, the reconstructive surgeon must have ample lead time prior to the operation. In order to design and engineer customized titanium plates, the reconstructive surgeon must obtain the necessary imaging studies and complete a planning session with the resecting surgeon and engineer with sufficient time to generate the guides and plate. Imaging studies also represent a financial cost to the healthcare system, additional radiation exposure and the use of contrast can be damaging to the kidneys, and time off work for the patient is necessary to complete the studies. In terms of timing, in our experience, generating a model and cutting guides can be performed with the shortest lead time; however, if a milled plate or 3D-printed plate is needed, the plan has to be verified one to two weeks, respectively, prior to the surgery date.

Another factor that is crucial to consider is the difficulty of modifying the plan after the models and plate have been manufactured. Most resections can be performed with adequate margins based on preoperative imaging; however, in the case that the cancer progresses and a more aggressive resection is needed, modification of the cutting guides and titanium plate can be extremely difficult and may require a free-hand approach if the guides and plate are rendered unusable. Under these circumstances, the operative time may be prolonged if the plans are abandoned, but the costs of the model and plate are wasted. This underscores the importance of careful planning and communication with the resecting surgeons to minimize the potential for altering the plan after the plan has been confirmed and production begins.

Despite the reported advantages of VSP delineated in the present review, some other additional limitations also need to be considered. In light of the potential for long-term complications such as hardware infection or exposure, consideration should also be given to the use of vascularized bone flaps. However, as the technology improves with imaging and medical modeling, other osteocutaneous flap donor sites aside from the fibula can also be designed with the same precision and accuracy as with the scapula or iliac crest or potentially even a bioengineered 3D-printed scaffold [47]. Nowadays, at many institutions, the use of VSP has quickly emerged as the gold standard and largely replaced traditional means of reconstruction, which are predominantly reserved for emergent cases when there is not sufficient time to generate the plans, cutting guides, and printed plates. Aside from the lead time, the costs of the models and especially the customized plates can be prohibitive. In order to address the costs and lead time needed for planning and processing, some teams have even developed their own in-house VSP systems and demonstrated excellent promising outcomes with their technology [48,49]. In other institutions, to compensate for the need for time to plan and design the models, a staged approach has been proposed to allow for precise reconstruction of the exact defect without compromising the oncologic treatment and delaying adjuvant therapies [50].

## 3. Conclusions

Microvascular reconstruction of the midface presents formidable challenges to the reconstructive surgeon; however, these challenges also offer tremendous opportunity for innovation and advances to optimize patient outcomes and reduce complications. The growth in virtual surgical planning has ushered in a new era in orbitomaxillectomy reconstruction with customized cutting guides and patient-specific 3D-printed titanium plates, but the reconstructive surgeon must also consider the limitations and costs associated with this evolving technology. Alternate donor sites and the use of vascularized osseous or osteocutaneous flaps can limit the need for hardware in certain defects and should be included in the armamentarium of microsurgeons performing a high volume of head and neck reconstructions.

## Figures and Tables

**Figure 1 medicina-59-01762-f001:**
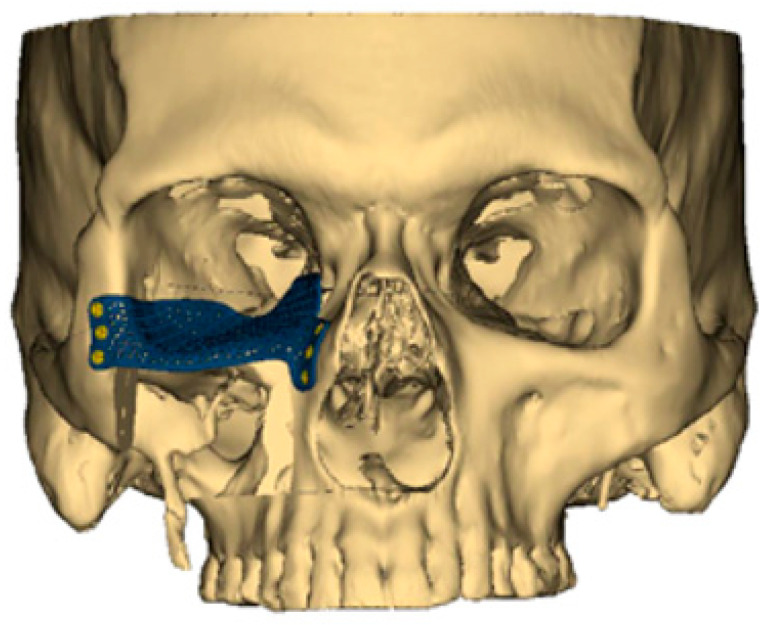
A customized 3D-printed titanium flap used for reconstruction of an orbital floor defect.

**Figure 2 medicina-59-01762-f002:**
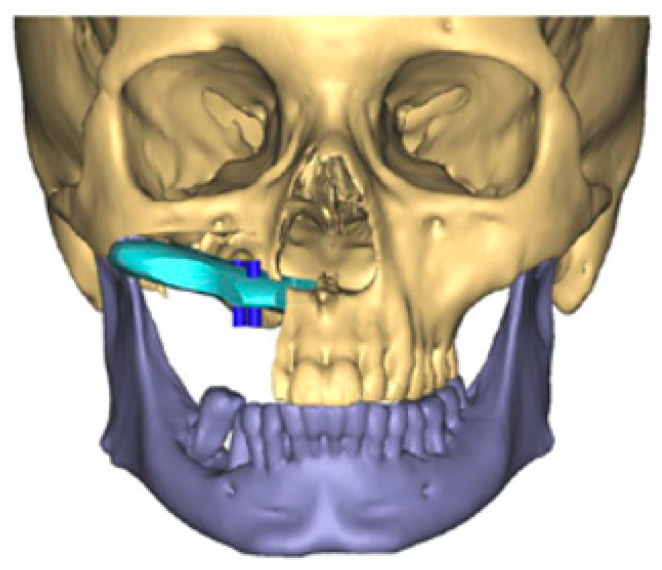
Three-dimensional reconstruction of computed tomography images of a maxillectomy reconstruction using medical modeling and virtual surgical planning. A scapula flap was used to reconstruct the maxilla with planning for immediate dental implants. “A scapula flap (light blue) was used to reconstruct the maxilla with planning for immediate dental implants (dark blue)”.

**Figure 3 medicina-59-01762-f003:**
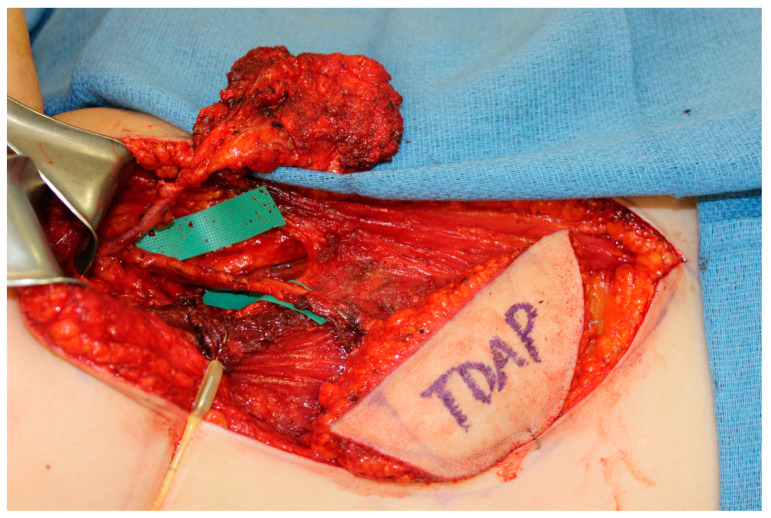
A chimeric scapula flap with a soft tissue thoracodorsal artery perforator (TDAP) flap. The bony component was utilized for the maxilla and the TDAP flap was used to reconstruct the palate.

**Figure 4 medicina-59-01762-f004:**
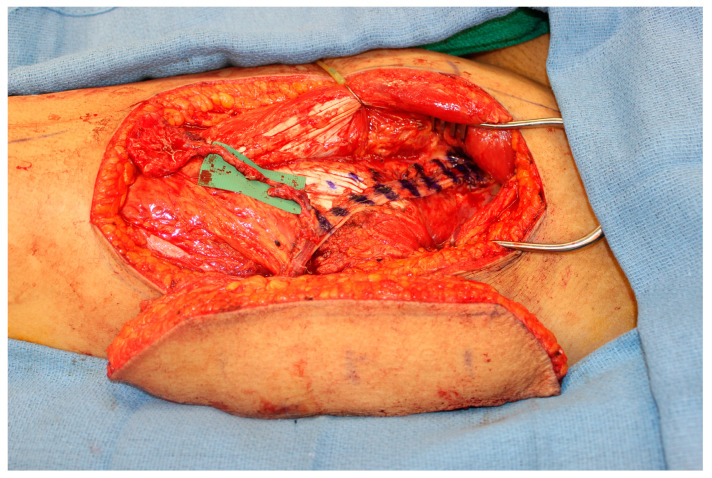
An engineered chimeric flap using the distal lateral descending circumflex femoral vessels on an ALT flap as the recipient vessels for a flow-through MFC flap. The pedicle of the ALT flap provides the length necessary to reach recipient vessels and also the soft tissue can be used to obliterate the dead space in the maxillary sinus or to reconstruct the palate.

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
