# Peer review of "Evolution of Medical Modeling and 3D Printing in Microvascular Midface Reconstruction: Literature Review and Experience at MD Anderson Cancer Center"

_medicina, 2023, doi:10.3390/medicina59101762_

Round 1
Reviewer 1 Report
The study is so valuable in promoting 3D Printing in Microvascular Midface Reconstruction. The research design is appropriate. However, there are some main comments.
The introduction section was not well-written. It dose not provide sufficient background and does not include all relevant references.
More recent and relevant references are needed to use.
How the patients were chosen?
How was the blinding?
How was the randomizing?
What were the statistical methods? software?
What were the references or standards for the tests?
The conclusion is so brief and does not support the results. please improve.
Minor editing of English language required
Reviewer 2 Report
The authors of the publication submitted for review are specialising in the field of plastic surgery. The paper focuses on reconstructive procedures of the midface with the use of VSP (virtual surgical planning) and presents three clinical cases of patients treated with this method.
In the present day and age, the implementation of tools such as VSP, resection templates and PSI (patient-specific implants) in microvascular midface reconstruction are crucial parts of effective treatment leading to even better functional and aesthetic results. The rising popularity of said surgical techniques requires constant development in that area e.g.with the means of publishing scientific research on the subject. A reliable empirical data on this topic is necessary due to its importance and constant need for scientific exploration. Therefore, I have to admit that the issues presented in this manuscript are very much up-to-date and interesting which I highly approve of.
The article has a rather unusual structure in terms of chapters – there is an Introduction, Case Presentation, Discussion and Conclusion. What is also not typical are the contents of those chapters. It is hard for me to assess whether this paper is a literature review or not, if it is though, I would suggest looking into the specific guidelines for this type of publication and make proper changes to the article accordingly:
https://guides.mclibrary.duke.edu/sysreview/types
Clearly pointing out what type of literature review has been done by the authors is extremely important to all potential readers, therefore this information should be included both in the title of the manuscript as well as the text itself. The selection methodology should also be described in detail regarding choosing the publications referenced in the manuscript, inclusion and exclusion criteria, types of databases searched, keywords, dates of login, etc. All of that data impact the scientific value and reliability of the article significantly, which means that the information mentioned above needs to be a part of the final manuscript.
However, if the authors’ intention was to present their research as a case report (three patient cases), this should also be clearly stated in both the title and contents of the article. If the manuscript is indeed a case report then it has to be edited in a way that is suitable for this type of publication following the guidelines.
Despite the fact that the authors mention relatively new an innovative methods of treatment, they should also prove what aspect of their research makes it unique and standout among other similar publications. What does this manuscript bring into the world of medicine and science? What findings or the way of presenting them were not known to the wider audience prior?
In my professional opinion, taking into account all the remarks and concerns I have regarding this manuscript, it is not eligible for publication in its current form. It requires major revision and extensive editorial work in order to be suitable to appear in any periodical or digital repository.
Round 2
Reviewer 2 Report
I am highly dissatisfied with the authors' response to my, and for that matter also other reviewers’, comments and suggestions. The changes made to the manuscript are mainly cosmetic and the text does not significantly differ from the previous version. Moreover, the adjustments that have been made are very superficial and do not go in accordance with my and other reviewers’ remarks. Instead, the authors are trying to get their point across and convince us that this paper is pretty much flawless. I wanted to remind the authors that it is in their best interest to carefully read and follow the advice given by the reviewers, especially when somehow all of them have pointed out similar issues with the manuscript.
Such an attitude may result in difficulties with releasing future articles unless there is a clear understanding that cooperation between reviewers, editors and authors is a crucial part of the publishing process and cannot be treated lightly. It is not in our interest nor is it our ill will to reject articles without valid reasons. On the contrary, the suggestions that all reviewers provide are there to aid the authors and help them improve their work, not cause frustration or feelings of injustice. Therefore, the article is not acceptable for publication in the current form and my stance will not change unless a major revision is made.
Author Response
Please see attached explanations
